# “Living an Obstacle Course”: A Qualitative Study Examining the Experiences of Caregivers of Children with Rett Syndrome

**DOI:** 10.3390/ijerph16010041

**Published:** 2018-12-25

**Authors:** Domingo Palacios-Ceña, Pilar Famoso-Pérez, Jaime Salom-Moreno, Pilar Carrasco-Garrido, Jorge Pérez-Corrales, Paula Paras-Bravo, Javier Güeita-Rodriguez

**Affiliations:** 1Department of Physical Therapy, Occupational Therapy, Rehabilitation and Physical Medicine, Universidad Rey Juan Carlos, Alcorcón, 28922 Madrid, Spain; domingo.palacios@urjc.es (D.P.-C.); Jorge.perez@urjc.es (J.P.-C.); 2Department of Nursing, Servicio Madrileño de Salud, 28004 Madrid, Spain; pilarfamoso50@gmail.com; 3Department of Physiotherapy, Universidad Francisco Vitoria, 28223 Madrid, Spain; jaime.salom@ufv.es; 4Department of Medicine and Surgery, Psychology, Preventive Medicine and Public Health, Inmunology and Microbiology Medicine, Nursing and Stomatology, Universidad Rey Juan Carlos, Alcorcón, 28922 Madrid, Spain; pilar.carrasco@urjc.es; 5Department of Nursing, Universidad de Cantabria, 39005 Santander, Spain; paula.paras@unican.es

**Keywords:** rare disease, Rett syndrome, caregivers, qualitative research

## Abstract

*Background*: Rett syndrome has considerable effects on the quality of life of affected children, impairing everyday activities and potentially impacting the life of both the caregivers and the family. Our aim was to explore the experiences of a group of caregivers of children with Rett syndrome with regards to living and caring for their children. *Methods*: We conducted a qualitative case study to examine how 31 caregivers of children with Rett syndrome perceived living with their children. Data were collected through in-depth interviews, focus groups, researchers’ field notes and caregivers’ personal documents. A thematic analysis was performed following the Consolidated Criteria for Reporting Qualitative Research (COREQ) guideline. *Results*: The experience of being a caregiver of a child with Rett syndrome was expressed as being akin to an “obstacle course”, and was described via three main themes: (a) looking for answers, with two subthemes identified, namely ‘the first symptoms’, and ‘the need for a diagnosis’; (b) managing day to day life, with the subthemes ‘applying treatments’, and ‘learning to care’; and (c) money matters. *Conclusions*: Rett syndrome has a considerable impact on the lives of the caregivers involved. The health-care process and the management of economic resources are some of the aspects highlighted by caregivers. These findings have important implications for the planning of support services, health systems and health policies.

## 1. Introduction

A neurological disorder, Rett syndrome (RTT) arises due to mutations in the X-linked gene methyl-CpG-binding protein 2 (*MECP2*), a ubiquitously expressed transcriptional regulator. It is a comparatively rare syndrome that occurs in around 1:10,000–15,000 live female births [1]. Rett syndrome can be identified via numerous symptoms. These include a decline in head growth, abnormalities of gait, loss of hand movements, which are invariably replaced with repetitive movement, loss of speech and respiration difficulties [1,2]. These functional problems can also be worsened by the development of comorbid conditions including gastrointestinal disorders, epilepsy and scoliosis [3,4]. 

The syndrome has been classified into three distinct presentations: typical, atypical, and variant [1,2]. Symptoms can generally be experienced between 6–18 months after normal pre- and postnatal development. These include severe intellectual disability along with the loss of acquired skills [5]. Furthermore, after a certain time span when normal neurological and physical development takes place, (generally during the first 6–18 months), early characteristics of RTT can appear which then develop over several stages: stagnation (age 6–18 months), rapid regression (age 1–4 years), pseudostationary (age 2–potentially life) and subsequent motor deterioration (age 10–life) [1].

This syndrome is highly complex and varies between individuals living with RTT, particularly with regards how the children experience social, physical, and communication challenges, which may restrict their social participation and activities, overall [3]. At present, there is no cure for RTT, rather the medical direction is aimed at providing symptomatic relief for patients based on a multidisciplinary approach [1,2]. Currently, neurobiologically-based drug trials are the ultimate goal in RTT. Due to the global nature and complexity of the disorder, drugs that target general mechanisms (e.g., growth factors) as well as particular systems (e.g., glutamate modulators) could be deemed successful [6]. Also, health and disability services for RTT include developmental therapies, based on gross motor function, and medical management of comorbidities, although, until now, positive results related to these interventions are low, overall [7]. 

Standardized questionnaires have shown that RTT can lead to a reduction in the quality of life (QOL) of children [3]. At the same time, the actual life experience and impact of RTT are extremely personal and differ within each family, with aspects that can never be accurately measured by formal scales or questionnaires [4]. This is because this is a process of informal childcare and a learning curve is involved, especially regarding the medical recommendations for caregivers to take care of their children. Moreover, the absence of resources influences the daily life of the family and impacts upon care for their children with RTT.

The experience and individual perspective of the caregivers concerned is relevant, just as one’s life experiences, ambitions and emotional needs should be considered when treating children with RTT [3,8,9,10,11,12,13]. Therefore, several questions remain unanswered: how do caregivers live and care for their children with RTT? What is their daily living experience? Thus, the purpose of this study was to explore the experiences of a group of caregivers of children with RTT and to understand what it is like living and caring for a child with RTT. The importance of this investigation is that it offers a description of the disorder and its impact from the point of view of caregivers. This study focuses on aspects such as the search for a diagnosis, the manner families care for their children on a daily basis, and how the economic aspects influence care and the life of the family and their children.

## 2. Materials and Methods

### 2.1. Design

A qualitative descriptive case study was conducted [14,15]. Qualitative methods are indicated to help understand the beliefs, values, and motivations that underlie individual health behaviors [16,17]. This type of case study is used to describe an intervention or phenomenon and the real life context in which it occurred [15]. In this study, the phenomenon under study is the impact of the disorder on the caregivers of children with RTT. 

### 2.2. Research Team

Prior to the study, the positioning of the researchers was established via two briefing sessions according to the theoretical framework, their beliefs, and their motivation for participating in the research [16,17]. The results of those sessions are shown in Table 1.

Seven researchers (four men and three women) participated in this study, including an occupational therapist (J.P.-C.), a pharmacologist (P.C.-G.), two physiotherapists (J.G.-R., J.S.-M.), a Registered Nurse (P.F.-P.) and two research nurses (D.P.-C., P.P.-B.). Four of the participants (D.P.-C., P.P.-B., J.P.-C., J.G.-R.) had experience in qualitative study designs, were not involved in clinical activity, and had no prior relation with the patients included. 

### 2.3. Context

Rett syndrome is a rare disorder [18,19]. Rare disorders are characterized by having an average prevalence of between 40 and 50 cases per 100,000 people [18,19]. Family associations play an important role in supporting families, providing true and up-to-date information, collaborating in care for the children and financing research [20]. The participants were recruited from the Mi Princesa Rett Association (https://miprincesarett.es/) and the Spanish Rett Syndrome Association (http://www.rett.es/).

### 2.4. Participants

Inclusion criteria: (a) Caregivers who, at the time of the study had children diagnosed with RTT, and/or legal guardian; (b) the diagnosis of RTT was made by the pediatrician and/or the neurologist, (c) children could present any variation of RTT, and (d) signing the informed consent. Exclusion criteria: (a) a diagnosis of RTT, not confirmed by the pediatrician and/or neurologist, and (b) not signing the informed consent. The RTT diagnosis performed by the neurologist and pediatricians was confirmed by consulting medical reports provided by the caregivers.

### 2.5. Sampling Strategies

A purposive, critical-case sequential sampling, based on relevance to the research question (not clinical representativeness), was used (20 participants) [21]. Also, a snow-ball technique was incorporated whereby the researcher identifies participants with experience in the topic of interest and asks whether they know other caregivers who might meet the inclusion criteria and be interested in participating in the study [16]. This led to 11 further participants being involved in the study. 

Sampling and data collection was continued until the researchers achieved information saturation, at which point no new information emerged from the data analysis [16,17]; in our study this situation occurred after the inclusion of 31 caregivers. 

### 2.6. Recruitment

The recruitment period took place throughout March, 2016. Researchers were introduced to the caregivers via the directors of both the participating associations. Thereafter, the researchers explained the purpose and design of the study to the individuals who met the inclusion criteria during an initial face-to-face contact session. A one week period was then allowed for patients to decide whether or not they wished to participate and they were given a copy of the informed consent for them to review. In a second face-to-face session, they were asked to provide written informed consent and permission to tape the interviews. All the selected caregivers agreed to participate in the study.

### 2.7. Data Collection

Data were collected over a seven-month period between April and October 2016. The objective of this case study was to obtain an in-depth multi-perspective holistic enquiry regarding the phenomena of interest, entailing the need for multiple data collection tools [22]. The first stage of data collection consisted of semi-structured interviews based on a question guide (Table 2), in order to obtain information regarding specific topics of interest [16,17]. Thereafter, the researchers listened carefully, noted the key words and topics identified in the caregivers’ responses and used their answers to request further clarifications [16]. In this way, relevant information was collected from the caregivers’ perspective. The question guide was developed based on a prior literature review [6,7,8,9,10,11,12,13] and the researchers’ experience [16].

The second stage consisted of a focus group (FG) based on a question guide (Table 2), in order to examine different perspectives within the same group, acquire understanding of the problems faced by the group and facilitate the identification of values and norms [16,23]. The FGs were conducted by a moderator following a uniform structure [16,23]. The question guide was sufficiently focused to gather information on the area of study, but open enough to stimulate discussion and interaction between the participants [23]. 

The interviews and FG were tape-recorded and transcribed verbatim. A total of 19 interviews were undertaken, and two FG. Each FG comprised seven and five participants, respectively. Overall, 1333 min of data collection were recorded, with 1073 min corresponding to the first stage and 260 min to the second stage. Each of the first stage interviews lasted between 73 and 183 min, while the second stage FG lasted between 96 and 164 min. All interviews were held at the respective associations or at the caregivers’ home, depending on the caregivers’ preference. Researcher field notes and personal letters provided by participants were also collected. Personal letters were used as a secondary source of information, to provide more in-depth information and support the data obtained from other data collection tools such as the interviews, [16]. All the participants were asked to voluntarily gather their experience in a personal letter, however, only only one personal letter was obtained from the caregivers, together with 21 researcher field notes. 

### 2.8. Data Analysis

A thematic, inductive analysis was performed [16,24]. This method of analysis is congruent with the design of the case study [22]. Complete literal transcriptions were made for each of the FGs, in-depth interviews, researchers’ field notes, and for the participants’ documents [17]. The thematic analysis approach [16,17,24] involved identifying the most descriptive content in order to convert the data into meaningful units, and subsequently reduce and identify the most common meaningful groups. In this manner, clusters of meaningful units were generated, i.e. similar points or content that allowed the emergence of the topics that described the study participants’ experience [16,17]. This thematic analysis procedure was used separately with the interviews, FGs, and with regards the personal documents. Joint meetings were subsequently held to combine the results of the analysis and to discuss the data collection and analysis procedures. In the event of differences in opinion, theme identification was performed based on a consensus among the research team members. Subsequently, the research team held joint meetings to show, combine, integrate and identify final themes [22]. No data analysis software was used.

### 2.9. Rigor 

The Consolidated Criteria for Reporting Qualitative Research guidelines [25] (http://www.equator-network.org/) and the recommendations for the design of Case Study Research in health care using the DESCARTE model [22] were followed. Furthermore, the criteria by Guba and Lincoln (Table 3) was used for establishing the trustworthiness of the data by reviewing issues concerning data credibility, transferability, dependability, and confirmability [16,26]. These methods to increase rigor are compatible with case-study designs [15,27]. Regarding the use of triangulation methods (researcher triangulation, participant triangulation, and data collection methods triangulation) (see Table 3) in this study, these sought to provide greater depth to the data and confirm the credibility of the data obtained and the interpretation performed by the researchers [16].

### 2.10. Ethical Considerations

All subjects gave their informed consent for inclusion before they participated in the study. The study was conducted in accordance with the Declaration of Helsinki, and the protocol was approved by the Clinical Research Ethics Committee at the Rey Juan Carlos University (Code: 220220161516).

## 3. Results

The study took place between February 25, 2016 and February 10, 2017. Thirty-one caregivers (17 women, 14 men), with a mean age of 45.38 (*SD* ± 10.85) years were included in the study. In total 83.9% (*n* = 26) were married, 6.4% (*n* = 2), were widows, and 9.7% (*n* = 3) were separated. The average age of RTT diagnosis for children was 4.50 (*SD* ± 3.56) years. The clinical and demographic features of participants are shown in Appendix A. The experience of being a caregiver of a child with RTT was experienced as being akin to facing an “obstacle course”, described via three main themes: (a) looking for answers, with two subthemes identified: the first symptoms, and the need for a diagnosis; (b) managing day-to-day life, with two subthemes applying treatments, and learning to care; and (c) money matters. Appendix A, reports some of the patients’ narratives taken directly from the interviews, focus groups and personal letters regarding the three emerging themes. 

### 3.1. Theme 1: Looking for Answers

This theme describes how the caregivers sought for explanations from the appearance of the first symptoms, how they faced these changes in their children and how they sought a diagnostic confirmation, in order to respond to all their questions, doubts and fears.

#### 3.1.1. Subtheme: The First Symptoms

The caregivers described how suddenly they began to notice changes in their children, which they had not seen before. The first symptoms were manifold, arising at irregular intervals, leading to a state of alert in the caregivers, which affected their initial feelings of joy after giving birth. Some caregivers described having difficulties perceiving that something was happening, it was as if they were blind despite the fact that it was occurring before their very eyes: “*You can’t believe it, everything happened in front of your eyes, and I didn’t realize a thing. It’s frustrating how you can trick yourself, it was like being blind.*” *(P2*, *33 years old).* Another relevant aspect in the narratives of the caregivers was the person who identified the symptoms. Among most participants, it was the mothers who were the first to realize that something was not right regarding the development of their child. If they had other children they were well aware of the development process of a newborn, or they compared their child with others and noticed differences. The caregivers described how the nursery school was often the place where early detection took place: “*Speaking with other parents, they say the same thing. When you take your daughter to the nursery school is when it all began. When they called me I already knew that something wasn’t right*” *(P18*, *36 years old).*

#### 3.1.2. Subtheme: The Need for a Diagnosis

The caregivers described how they felt the need for a diagnostic confirmation of their suspicions and fears as early as possible in order to avoid delaying treatment. Because of the lack of diagnosis or the delays, the families often sought information from specialist doctors, initially avoiding an internet search: “*I am criticized for searching the internet, but they don’t realize, we need up-to-date information quickly, we can’t wait. Waiting means delaying treatment. The diagnosis is a priority.*” *(P21*, *55 years old).* Furthermore, the caregivers spoke of the large number of tests they had to undergo, which further delayed the diagnosis. All the participants remarked on the importance of a genetic diagnosis as being the necessary and essential test. The families described how the illness in males is rare, therefore being a boy is a factor that may hamper the petition to perform genetic tests. In general, the search for a genetic diagnosis is experienced as a pilgrimage around different differential diagnoses. However, the multitude of diagnostic tests was the major cause of burnout for caregivers: “*What causes the most burnout is doing tests and knowing that there is no response, but you have to do something, right? But it causes a great deal of burnout.*” *(P4*, *45 years old).* The caregivers described how some tests were painful, while others were costly and had to be covered by other family members to speed up the process. Furthermore, the costs that were covered by the public health system were often delayed due to the lengthy procedures and waiting times involved. 

### 3.2. Theme 2: Managing Day to Day Life 

This theme focuses on how caregivers attempt to integrate the treatments and care into their daily life and the family routine.

#### 3.2.1. Subtheme: Applying Treatments

This subtheme describes the medical pharmacological and non-pharmacological treatments received (physiotherapy, speech therapy). The caregivers related that they had to decide on the type of therapy to give their child, considering the accessibility of the same regarding available timetables, distance from their home and cost.

Medical check-ups condition the plans of the family and caregivers, as the whole family must adapt to the medical timetables, modifying their routines and availability at work: “*What’s important is the doctors and the check-ups, sometimes it isn’t fair for the whole family, but that’s the way things are.*” *(P5*, *48 years old).* For the caregivers, the administration of medication is, at times, a great obstacle, due to the illness and its symptoms, and the caregivers must develop strategies and tricks to administer the same. On the other hand, not being able to administer the medication is a considerable source of stress for them as it means that symptoms are not controlled as expected which may mean the child’s condition worsens: “*Not being able to give him his treatment or doing something that they tell you, is a source of continuous stress, especially because you can’t accept him getting worse because you haven’t been able to administer the medication, for example.*” *(P18*, *36 years old).* The caregivers describe not being able to make do without non-pharmacological therapies, such as speech therapy or physiotherapy. For them, maintaining these therapies means fighting against the illness on another front, as they attempt to decrease the physical and cognitive disability of the child. Physiotherapy appears in the narratives as being a basic pillar for everyday work, however, there are considerable difficulties associated with receiving continuous physiotherapy treatment in the public health system, as treatments are generally of a short duration and, after discharge, private physiotherapy services are expensive: “*They are amazing, and you can tell when we go to the physiotherapist. If it were up to me I would go every day, and I would stop doing many things, but it’s too expensive and in the public system you have a limited number of sessions in time.*” *P31*, *25 years old).* Also, there are other therapies, such as aquatic therapy, therapy with animals, music therapy, speech therapy and occupational therapy. Many caregivers, upon deciding the type of therapy they prefer, attempt to offer the child activities that also prevent decline, serve as a stimulus and enable the child to enjoy the activity. 

#### 3.2.2. Subtheme: Learning to Care 

Caregivers describe how they must learn to care for their children and face ever-changing problems without a fixed pattern, these are problems derived from the illness, such as: nutritional problems, bowel problems, sleeping disorders, respiratory disorders, epileptic seizures and stereotypical movements: “*Suddenly, one day you wake up and you have to do and know about everything, you need to know about medicine, nursing, occupational therapy, nutrition…*” *(P19*, *39 years old).* One of the main concerns of caregivers is the risk of malnutrition, dehydration and infection, due to the presence of vomiting, gastroesophagic reflux, episodes of broncoaspiration and pain. Thus, the caregivers must learn to prepare meals with nutritional ingredients that are easy to swallow, as well as controlling episodes of nausea and vomiting. Together with nutrition, bowel and bladder elimination is perceived as being one of the pillars of care for the children, which forces the need for caregivers to learn the management of urinary incontinence, the administration of laxatives and enemas and how to perform manual evacuation. On the other hand, sleeping difficulties have a major impact on the daily life of the whole family. This aspect generates a great amount of stress and anxiety on the caregivers as they feel that when they are tired they are unable to care for their child appropriately: “*As a father, you would like to be able to help, and especially resolve everything and protect your daughter from so many bad things… But the illness makes you see that it isn’t so [not being able to protect your child] and this causes tremendous anxiety and frustration.*” *(P17*, *41 years old).* Additionally, epileptic seizures are feared and met with anxiety and stress. The caregivers are forced to manage these wherever they may take place. To deal with this, the main strategy used by caregivers is that of constant vigilance, making some caregivers feel overprotective at times. Some caregivers relate the presence of epileptic seizures with the arrival of menstruation, causing frequent gynecological and menstrual checkups as tools to prevent the crises. Despite the fact that the stereotypical movements are a characteristic symptom of RTT, this was unknown by most participants. The families attempted to manage and decrease these with different strategies such as the use of splints and clothing, yet without success, meaning that the sucking of the hand eventually led to skin breakdown and/or damage. The uncontrolled movements are described as being distressing situations that, at times, they are unable to contain and which force them to modify the house to avoid self-injury.

### 3.3. Theme: Money Matters

Rett syndrome causes a high impact on a family’s financial resources. Families must cover all expenses not directly related with hospital health care. In addition, there are work readjustments that the couple must make in order to take care of their child, in terms of a reduction in working hours, thus decreasing their economic income: “*This illness is not just a medical problem, the money is essential, not only are there medical difficulties, money is always there, and that’s where it limits your access to services and/or therapies. Even when you are ill there are differences if you have money…*” *(P6*, *38 years old).* The families must cover expenses for treatments that are not covered on public healthcare, such as physiotherapy, speech therapy and occupational therapy, and this represents an indispensable monthly expense. On the other hand, the cost of the sanitary supplies that the caregivers must cover is considered abusive. There are no regulations and they feel exploited because of their needs: “*The government should do something to control it, you not only have to face seeing your daughter ill, you have to face feeling exploited because of your misfortune, everything costs money and sometimes with no regulation, I feel they cheat us and they take advantage of our need to take care of our daughters.*” *(P21*, *55 years old)*. Likewise, to improve the comfort, accessibility and quality of life of the children, the home and car must be adapted to the changing needs caused by the illness. This is an additional cost which the families describe as being constant and never-ending. Furthermore, the caregivers perceive a great inequality for accessing the genetic diagnosis according to the city where they live, as there are cities where the public health services cover this diagnosis, while in others it is the families themselves who must cover the cost of the diagnosis, leading to situations of anguish as they lack the sufficient resources. Among the solutions, some families managed to be included in research projects which covered the cost of the tests, whereas others had to request a loan from the bank or buy second-hand material. This leads to major disparities between the families regarding the treatment they receive and the material resources at hand. In this situation, there is a strong solidarity movement among the families where materials and resources are donated and shared: “*When you have an illness like this one, you can’t avoid feeling lonely, it is your daughter who suffers it, everything is difficult, and suddenly you meet people with the same difficulties who help you without asking for anything in exchange, just because they have gone through what you have experienced, or they are still going through. For us other families, the solidarity is amazing, it’s something you can’t even begin to explain.*” *(P12*, *49 years old).*

## 4. Discussion

Caregivers of children with RTT experience the illness as being like an “obstacle course”, where they must continuously overcome hurdles. These include hindrances for finding responses to their symptoms and achieving a diagnosis, for managing the treatment and daily care, and for finding the essential financial resources to meet all the expenses generated by the illness. 

The way the family faces the obstacles and difficulties is determined by family functioning, a dynamic phenomenon that has been described as a family’s ability to achieve goals integral to the lives of its members [28]. Family functioning is a mediator which facilitates caregivers’ adaptation, and which acts upon the expectations of caregivers and their coping strategies [28]. These authors [28] identify that those families who do not share the responsibility and burden of caring for the children, and where all the workload is assumed by just one member of the family/couple, and when the symptoms have appeared late, is predictive of a low family functioning. Furthermore, these families will have a poorer sense of adaptation and lower potential in finding effective strategies for facing the daily challenges.

Our results show how the mothers are those who identify the first symptoms of RTT, coinciding with previous studies [3,13,28,29], where the person who discovered the first symptoms was the primary caregiver, a role which was faced by the women [3,11,13,28]. These symptoms can appear in an irregular manner, although they show a continuous progression which forces the mothers to seek information and help from the health professionals [29]. During this period, they attempt to explain the behaviors which are not “normal” in the development of the child, comparing these with their interactions with other children [13].

Our findings highlight the difficulty encountered by many caregivers in terms of identifying symptoms of alarm in their children. This may be explained due to the fact that, during the first phase of RTT, from birth to 6 months, the child may not show any manifestation of the illness [1,30]. Subsequently, during the second stage (regression stage), between 6–18 months, different symptoms arise, affecting speech, walking ability, socially withdrawn behavior and stereotypical hand movement, as well as sleeping and eating, epilepsy and respiratory problems [1,30]. For this reason, caregivers did not observe different behaviors in the various stages of RTT and possibly could not identify some early symptoms [30].

This study shows how the caregivers seek for responses to the symptoms via a diagnostic confirmation. Previous studies [29,31] reported that, during this process, the families suffer from feelings of worry, stress, frustration, sadness, doubt, exhaustion and fear of social rejection. Furthermore, the caregivers, in this search for a diagnosis, present a greater state of vulnerability, fragility and despair [19].

The experience of our participants regarding “being on a pilgrimage” in search of a diagnosis is a common one and described in previous studies [13,19,29]. Lopes et al. [19] show how this pilgrimage takes place among various health institutions and specialists, due to the lack of structured health policies and specialized reference centers for rare disorders. Furthermore, some professionals describe how the specialists in primary care, as well as specialist doctors experience discomfort because the system is slow and inefficient for resolving the needs of the caregivers and children [19]. Knott et al. [29], describe the testimony of a mother, narrating how she experienced a lack of knowledge on behalf of the health professionals, the fact that her daughter suffered from a multitude of tests without conclusive results, until reaching the *MECP2* genetic test. It is interesting how the genetic diagnosis is key to finding help. This can be explained because, in order to receive public aid, in some countries, it is necessary to have an “exact” diagnosis of the illness, and not just a “suspected diagnosis” or “syndrome”.

Our results do not include narratives regarding health professionals. This contrasts with other studies [13,19,29], in which the caregivers perceived a lack of training of professionals, and an absence of experience in the treatment of RTT, or, on occasion, a stigma towards the people with disability. On the other hand, Lopes et al. [19] point out that the scientific and technological resources cannot be effective if these are not available to health professionals during their academic training or via continuous medical education.

These findings show that the continuous need for medical check-ups forces the family to modify their routines. Maintaining daily routines provides a feeling of security for the child who is thus able to regularly participate in programmed family activities, with people they know and in a recognizable context [3].

The caregivers in our study thought highly of the pharmacological treatment and its administration in order to control the symptoms. This differs from the results of Lim et al. [12,13], where they describe how the caregivers feel a lack of hope and confidence regarding its effectiveness (especially regarding antiepileptic medication) and regarding its safety as these can produce secondary effects with prolonged use. However, the same study shows how caregivers are unsure of whether reducing the dosage or withdrawing the medication is a good option, as the symptoms can worsen or reappear.

In this study, the caregivers attempted to apply other therapies to stimulate their children. This is in line with Epstein et al. [3] who describes how contact with the natural environment outside the home and participation in a variety of outdoor activities is key for the quality of life of children with RTT. Thus, the children spend time with animals, explore new places, perceive the sensation of different elements (earth, water), and different weather (rain, wind, sun).

Our results show how, during the wait and/or after the diagnosis, the caregivers try to manage all the treatments and medical recommendations as well as providing the best care for their child. 

Because RTT is unexpected, the caregivers must face stressful situations and behavior, such as shouting and inconsolable wails, sleep disorders, eating and orthopedic problems, and difficulties with communication [4,31]. The families must reorganize their life and focus on a child with physical, cognitive and communication difficulties [12,13,31]. Caring for a child with RTT therefore requires multiple skills which, at times, may overwhelm the caregivers, along with having to face the associated financial demands, obliging them to maintain their jobs and advance in their careers [13,31]. Despite the fact that previous studies [28,32,33] report how caring and raising girls with RTT can negatively affect the quality of life of caregivers, and decrease satisfaction with their marital relationships [33], in our study this issue was not voiced by participants. 

Regarding the management of symptoms, the caregivers in this study displayed great concern with regards to malnutrition, dehydration and infections. This coincides with previous studies [13,32] which also reflect concerns regarding the ability to walk and use the hands, as well as communication difficulties. Considerable attention is given to eating [32], which may be explained because of the fact that nutritional supplementation as a therapeutic option means the use of feeding routes via a nasogastric tube or a gastrostomy tube [4,13,34]. 

This study exposes the high economic costs that families must cover, so much that meeting the financial demands in order to help with the care and treatments in the short and medium term becomes a priority. This is in line with previous reports [12,13,29,31,32] which highlight the financial difficulties caregivers have for buying equipment, hiring professionals, as well as the additional financial burden the illness has on families, worsened by one of the caregivers abandoning their work in order to take care of their child. Previous studies [13,31,33] show that it is usually the woman who is forced to leave the workplace, however this option is not for all families, as this would mean a lower income for the family [13]. 

As a rare syndrome, RTT is forgotten in the development of new medicines based on clinical trials [7,19]. A legal battle exists for caregivers in order to access therapies that the health system and the pharmaceutical companies do not consider a priority [19]. Other struggles concern obtaining an early diagnosis, the development of a joint working relationship between the family and the professionals, and preparing the family to develop coping strategies and learning how to care for their child [7,29]. Clarke and Abdala Sheikh [7] wrote: “*If there is one lesson that the history of therapeutic discoveries has taught us, it is that throwing money at the problem is no guarantee of the outcome you hoped for. While that shouldn’t stop us from seeking a cure, perhaps we should not give up the search for therapies that bring benefit without necessarily achieving a cure.*” (P4, 45 years old). Ultimately, people with RTT must be able to take care of themselves, have a decent quality of life, and live a good life [11,12,29].

The limitations of our study include, first, the fact that these results cannot be extrapolated to all caregivers who have children with RTT, owing to the design used. The presence of different levels of severity in children could modify the results, due to the fact that caregivers experience different situations and difficulties depending on the severity of the disorder. This may affect aspects such as the management and application of daily care or the economic resource needs for the purchase of devices, among others. However, these results may help professionals understand the caregivers’ experience, which is important to appreciate the impact of RTT on caregivers’ daily life and their management of the disorder. These new investigations may help describe and explore other phenomena, such as the impact of rehabilitation therapies on children, the learning and self-care processes experienced by caregivers, and the process of adaptation that the caregivers undergo both at home and in the community.

## 5. Conclusions

This study provides insight into how RTT is experienced in a group of Spanish caregivers with children with the syndrome. Our findings shed light on how RTT may impact the lives of the caregivers of children with RTT with important implications for clinical practice. 

These data could be useful for informing how clinicians counsel families at diagnosis, drawing upon aspects of life, such as treatment and care administration, health problems management, and economic resources management. Understanding the caregivers’ experience may improve our awareness of the daily life situations of caregivers and the difficulties and obstacles that they must overcome when caring for their children. Our study provides a basis for further studies addressing the impact of RTT on the life of caregivers, as well as issues regarding quality of life and care processes in children with RTT.

## Figures and Tables

**Table 1 ijerph-16-00041-t001:** The positioning of the researchers.

**Theoretical framework**	Researchers based their approach on a constructivist paradigm. This paradigm was based on the assumption that human beings construct their own social reality, and that knowledge is built through increasingly nuanced reconstructions of individual or group experiences.
**Beliefs**	RTT is a syndrome that appears during the first year of life and which manifests as a regression in the neurodevelopmental process of a child. This can lead to a traumatic experience in family life. However, it is important to understand which are the most relevant aspects for the caregivers and which have a greatest impact on their life, from their perspective.
**Motivation for the research**	Understand the experiences of the caregivers and how they manage their day to day life and how they care for their children suffering from RTT. The scarce amount of qualitative international studies on this topic and the absence of publications in Spain, warrants the need for qualitative research that explores the caregivers’ perspective.

**Table 2 ijerph-16-00041-t002:** Questions guide.

Investigated Theme	Questions
Experience with the illness	What is your experience and perspective of Rett syndrome?
Detection of symptoms	How were the first symptoms identified? What aspect was most relevant for you?
Diagnostic tests and genetic diagnosis	How was the diagnostic process? How was the moment of the diagnosis?
Impact on the family	How did the illness influence your family life and the relationship with the family members? What was most relevant for you? How did the family respond to news of the diagnosis?
Day-to-day life	How is your everyday life? What is most relevant for you? What was it like before and after the RTT diagnosis?
Daily process of care and management	How do you manage and care for your child with RETT? What is the most relevant aspect for you in your daily care duties?

**Table 3 ijerph-16-00041-t003:** Trustworthiness criteria.

Criteria	Techniques Performed and Application Procedures
Credibility	Investigator triangulation: each interview was analyzed by three researchers. Thereafter, team meetings were performed in which the analyses were compared and categories were identified. Participant triangulation: The study included caregivers with children of different ages, and who had undergone different stages of the illness. Thus, multiple perspectives were obtained with a common link (the experience of having a child with RTT). Triangulation of methods of data collection: semi-structured interviews and focus groups were conducted and researcher field notes were kept. Participant validation: this consisted of asking the participants to confirm the data obtained at the stages of data collection and analysis. All participants were offered the opportunity to review the audio or written records as well as the subsequent analysis to confirm the interpretation of their experience by the researchers.
Transferability	In-depth descriptions of the study performed, providing details of the characteristics of researchers, participants, contexts, sampling strategies, and the data collection and analysis procedures.
Dependability	Audit by an external researcher: an external researcher assessed the study research protocol, focusing on aspects concerning the methods applied and study design. Also, an external researcher specifically checked the description of the coding tree, the major themes, patients’ quotations, quotations’ identification, and theme descriptions.
Confirmability	Investigator triangulation, participant triangulation, and data collection triangulation. Researcher reflexivity was encouraged via the performance of reflexive reports and by describing the rationale behind the study.

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
