# Peer review of "“Living an Obstacle Course”: A Qualitative Study Examining the Experiences of Caregivers of Children with Rett Syndrome"

_ijerph, 2018, doi:10.3390/ijerph16010041_

Reviewer 1 Report

Dear Authors:

This is a noteworthy investigation and will contribute to our understanding of the impact of RTT on caregivers. There are some areas that require elaboration, but the overall study was well conducted and will be an important contribution to the field.

“Living an obstacle course”: the experience of parents of children with Rett syndrome: a qualitative study

Manuscript Review

Title: This may be more appropriate: “Living an obstacle course”: A qualitative study examining the experiences of caregivers of children with Rett syndrome

General: You may want to use the term caregivers instead of parents.

Abstract:
Line 23: Check grammar:  …with regards to living and caring…
Line 31: Conclusion: Instead of focusing on the beneficial use of qualitative research, discuss the overarching conclusion of the research investigation.

Introduction:
Line 64: Which aspects can never be accurately measured by formal questionnaires. Please elaborate on this.
Line 70. The aim of this study is important, so please elaborate on the importance of your investigation.

Materials and Methods
Be more specific regarding application of the Triangulation method.
Line 77: Be consistent with terminology. Instead of disease, use syndrome or disorder.
Line 126: Include a citation after that final sentence with reference to a prior literature review.
Line 101: Written and verbal consent? How long was the recruitment period?
Line 142: Elaborate on why personal letters were obtained. Why did only one parent submit a letter?

Results:

Line 196: What was the average age of diagnosis of RTT for the children? Any discussion of severity of RTT and its impact on the emerging themes?
Line 225: Were parents aware of evidence-based treatment options versus non-evidence-based treatment options? Did parents find therapy to be helpful on a personal front?

Discussion:

Lines 378-381: The strengths of the study do not need to be reiterated.
Line 382: Regarding limitations, elaborate on the severity of RTT among the children. You mention children were not interviewed due to intellectual disability, but please be more specific.

Conclusions:

Line 392: Again, refer to RTT as a syndrome or a disorder, and not as a disease.
Line 398: What is the issue with compliance that you are discussing here? Focus on the core findings of the study in this section.

Author Response

Reviewer 1

This is a noteworthy investigation and will contribute to our understanding of the impact of RTT on caregivers. There are some areas that require elaboration, but the overall study was well conducted and will be an important contribution to the field.

Response: we would like to thank the reviewer for this kind comment, and thank you for giving us the chance to elaborate further on our project, the comments received have been highly insightful and have enabled us to greatly improve the quality of the contents.

“Living an obstacle course”: the experience of parents of children with Rett syndrome: a qualitative study

Manuscript Review

Title: This may be more appropriate: “Living an obstacle course”: A qualitative study examining the experiences of caregivers of children with Rett syndrome

Response: According to the reviewer´s suggestion, we have revised and changed the title, thank you.

General: You may want to use the term caregivers instead of parents.

Response: thank you, according to reviewer´s suggestion we have revised and edited this term throughout the text.

Abstract: Line 23: Check grammar:  …with regards to living and caring…

Response: Edited, thank you

Line 31: Conclusion: Instead of focusing on the beneficial use of qualitative research, discuss the overarching conclusion of the research investigation.

Response: According to the reviewer´s suggestion, this paragraph has been edited. The current conclusion reads as follows:

Rett syndrome has a considerable impact on the lives of the caregivers involved. The health-care process and the management of economic resources are some of the aspects highlighted by caregivers. These findings have important implications for the planning of support services, health systems and health policies.

Introduction:

Line 64: Which aspects can never be accurately measured by formal questionnaires. Please elaborate on this.

Response: According to reviewer´s suggestion, this paragraph has been edited. The current text reads as follows:

At the same time, the actual life experience and impact of RTT are extremely personal and differ within each family, with aspects that can never be accurately measured by formal scales or questionnaires [4]. This is because this is a process of informal childcare and a learning curve is involved, especially regarding the medical recommendations for caregivers to take care of their children. Moreover, the absence of resources influences the daily life of the family and impacts upon care for their children with RTT.

Line 70. The aim of this study is important, so please elaborate on the importance of your investigation.

Response: thank you for this suggestion, we have added the following sentence to the text:

Therefore, several questions remain unanswered: how do parents live and care for their children with RTT? What is their daily living experience? Thus, the purpose of this study was to explore the experiences of a group of parents of children with RTT and to understand what it is like living and caring for a child with RTT. The importance of this investigation is that it offers a description of the disorder and its impact from the point of view of caregivers. This study focuses on aspects such as the search for a diagnosis, the manner families care for their children on a daily basis, and how the economic aspects influence care and the life of the family and their children.

Materials and Methods

Be more specific regarding application of the Triangulation method.

Response: thank you for pointing this out. We have included new text explaining why we applied triangulation as a technique. However, seeing as table 3 explains the types of triangulation used in this study we have been careful to not repeat the information already included in the table to avoid redundant text.

We included:

Furthermore, the criteria by Guba and Lincoln (Table 3) was used for establishing the trustworthiness of the data by reviewing issues concerning data credibility, transferability, dependability, and confirmability [16,26]. These methods to increase rigor are compatible with case-study designs [15,27]. Regarding the use of triangulation methods (researcher triangulation, participant triangulation, and data collection methods triangulation) (see table 3) in this study, these sought to provide greater depth to the data and confirm the credibility of the data obtained and the interpretation performed by the researchers [16].

Line 77: Be consistent with terminology. Instead of disease, use syndrome or disorder.

Response: We have replaced the term “disease” for “syndrome” or “disorder” throughout the text.

Line 126: Include a citation after that final sentence with reference to a prior literature review.

Response: this has been edited as follows:

The question guide was developed based on a prior literature review [6-13] and the researchers’ experience [16].

Line 101: Written and verbal consent? How long was the recruitment period?

Response: Written consent was obtained from participants:

To clarify this, we have included the following information in the recruitment section:

The recruitment period took place throughout March, 2016. Researchers were introduced to the parents via the directors of both the participating associations. Thereafter, the researchers explained the purpose and design of the study to the individuals who met the inclusion criteria during an initial face-to-face contact session. A one week period was then allowed for patients to decide whether or not they wished to participate and they were given a copy of the informed consent for them to review. In a second face-to-face session, they were asked to provide written informed consent and permission to tape the interviews. All the selected caregivers agreed to participate in the study.

Line 142: Elaborate on why personal letters were obtained. Why did only one parent submit a letter?

Response: The reason we used personal letters is because it represented a secondary source of information/data and support for other data collection tools such as interviews. We asked all parents to describe their experience, although only one parent handed it in. Among the reasons for not doing so were: not having enough time, or thinking that the information provided in the interviews and focus groups was sufficient. 

The following text has been included:

Personal letters were used as a secondary source of information, to provide more in-depth information and support the data obtained from other data collection tools such as the interviews, [16]. All the participants were asked to voluntarily gather their experience in a personal letter, however, only one personal letter was obtained from the parents, together with 21 researcher field notes.

Results:

Line 196: What was the average age of diagnosis of RTT for the children? Any discussion of severity of RTT and its impact on the emerging themes?

Response: To respond to this question, and clarify this for readers, we have added the following sentence:

The average age of RTT diagnosis for children was 4.50 (SD ± 3.56) years.

Regarding how severity of RTT can influence the results, we have added the text below to the limitations section:

The presence of different levels of severity in children could modify the results, due to the fact that caregivers experience different situations and difficulties depending on the severity of the disorder. This may affect aspects such as the management and application of daily care or the economic resource needs for the purchase of devices, among others.

Line 225: Were parents aware of evidence-based treatment options versus non-evidence-based treatment options? Did parents find therapy to be helpful on a personal front?

Response: We are unable to provide a clear-cut response to this question, as we did not ask parents openly, nor did the parents raise the issue of whether treatment was based on evidence or not. We can confirm that for the caregivers there was a difference between treatment with and without a confirmed Rett diagnosis, as once there is a diagnosis, the focus is on directing treatments towards the treatment of Rett.

Discussion:

Lines 378-381: The strengths of the study do not need to be reiterated.

Response: thank you, according to reviewer´s suggestion we have deleted the strengths of the study from the “discussion” section

Line 382: Regarding limitations, elaborate on the severity of RTT among the children. You mention children were not interviewed due to intellectual disability, but please be more specific.

Response:

Regarding how the severity of children can influence the results, we have included the text below in the limitations section:

The presence of different levels of severity in children could modify the results, due to the fact that parents experience different situations and difficulties depending on the severity of the disorder. This may affect aspects such as the management and application of daily care or the economic resource needs for the purchase of devices, among others.

On the other hand, the reason we did not include children was because we wanted to study the parents’ perspective. However, as researchers, in initial stages of the study we contemplated the possibility of including the joint perspective of children and parents. However, due to communication difficulties with the children and difficulties understanding some of the questions, we discarded this phase in the final study. Therefore, we believe it is necessary to remove the following sentence:

Also, children were excluded from the interviews due to their limited communication and cognitive skills.

Conclusions:

Line 392: Again, refer to RTT as a syndrome or a disorder, and not as a disease.

Response: thank you, we have revised and edited this concept for consistency

Line 398: What is the issue with compliance that you are discussing here? Focus on the core findings of the study in this section.

Response: thank you, we have revised and deleted this phrase and added a new one to make it clearer to the reader. The revised text is shown below:

These data could be useful for informing how clinicians counsel families at diagnosis, drawing upon aspects of life, such as treatment and care administration, health problems management, and economic resources management. Understanding the caregivers’ experience may improve our awareness of the daily life situations of caregivers and the difficulties and obstacles that they must overcome when caring for their children. Understanding the parents’ experiences may contribute to better communication between clinicians and parents and to a greater compliance with treatments. Our study provides a basis for further studies addressing the impact of RTT on the life of parents, as well as issues regarding quality of life and care processes in children with RTT.

Reviewer 2 Report

This qualitative study of parents of children with Rett syndrome (RTT) opens up about the “obstacles” one perceives when raising a child with RTT.  The study examines 31 parents’ views through the use of interviews, focus groups, researcher field notes, and personal documents.  This qualitative research gives insight into the way parents of children diagnosed with RTT perceive the illness and the impact it has on the QOL of both parents and children.

The completion of a study of this type is a much needed addition to the literature.  Far too often caregivers are overlooked in regards to the impact on life that disabilities have.  I have only a few comments.

Specific Comments:

Abstract:

Introduction:

Line 39: Insert a “,” for “1-10 000-15 000 live female births.”

Line 52: Consider replacing “sufferers” with “individuals living with RTT”.  In lines 62-67, you stated that every experience with RTT is different. My suggestion stems from the understanding and the need to change societal views on disabilities in that not all individuals have a life of suffering, but a life full of lived experiences that differ from one another.

Materials and Methods:

Line 98: What is meant by “legal tutors”?

Results:

None

Discussion:

Line 355: Insert “to” in the sentence “…displayed great concern with regards [to] malnutrition, dehydration and infections.”

Conclusions:

None

Author Response

Reviewer 2

This qualitative study of parents of children with Rett syndrome (RTT) opens up about the “obstacles” one perceives when raising a child with RTT.  The study examines 31 parents’ views through the use of interviews, focus groups, researcher field notes, and personal documents.  This qualitative research gives insight into the way parents of children diagnosed with RTT perceive the illness and the impact it has on the QOL of both parents and children.

The completion of a study of this type is a much needed addition to the literature.  Far too often caregivers are overlooked in regards to the impact on life that disabilities have.  I have only a few comments.

Response: we would like to thank the reviewer for his/her kind comments, and for giving us the chance to review and update our manuscript. The suggestions are highly insightful and have enabled us to greatly improve the quality of the contents.

Specific Comments:

Abstract:

Introduction:

Line 39: Insert a “,” for “1-10 000-15 000 live female births.”

Response: Edited, thank you

Line 52: Consider replacing “sufferers” with “individuals living with RTT”.  In lines 62-67, you stated that every experience with RTT is different. My suggestion stems from the understanding and the need to change societal views on disabilities in that not all individuals have a life of suffering, but a life full of lived experiences that differ from one another.

Response: thank you, we totally agree, we have revised and edited this concept for consistency

Materials and Methods:

Line 98: What is meant by “legal tutors”?

Response: This was a mistake, we have corrected the term, replacing it with legal guardian”.

Results:

None

Discussion:

Line 355: Insert “to” in the sentence “…displayed great concern with regards [to] malnutrition, dehydration and infections.”

Response: Edited, thank you

Conclusions:

None

Reviewer 3 Report

This is a well written and nicely designed qualitative study on the experiences of parents of children with Rett syndrom.

I only have a few suggestions to make:

How did you verify that the diagnosis was confirmed by a neurologist or pediatrician.

In table 3 you mention non-structured interviews in addition to semi-structured ones but in the Methods section only semi-structured interviews are introduced. Please clarify.

Please provide two quotations per topic in Table S2.

It would also be nice to provide some sample quotations in the main text as well.

Author Response

Reviewer 3

This is a well written and nicely designed qualitative study on the experiences of parents of children with Rett syndrom.

Response: thank you for your kind comment and for giving us the chance to review and update our manuscript. The comments have been highly insightful and have enabled us to greatly improve the quality of the contents.

I only have a few suggestions to make:

How did you verify that the diagnosis was confirmed by a neurologist or pediatrician.

Response. This was verified by consulting medical reports issued by neurologists and pediatricians and provided by the parents. We have clarified this point in the inclusion criteria: 

The RTT diagnosis performed by the neurologist and pediatricians was confirmed by consulting medical reports provided by the caregivers. 

In table 3 you mention non-structured interviews in addition to semi-structured ones but in the Methods section only semi-structured interviews are introduced. Please clarify.

Response: We agree with the reviewer, this has been modified as follows:

Triangulation of methods of data collection: non-structured interviews, semi-structured interviews and focus groups were conducted and researcher field notes were kept.

Please provide two quotations per topic in Table S2.

Response: We have made these changes as suggested by the reviewer (two quotations per topic in Table S2).

It would also be nice to provide some sample quotations in the main text as well.

Response: We have followed the reviewer’s recommendations. In order to avoid weighing down the text, having used table S2 as a source of narratives and including more narratives, we have included examples of quotations in the most relevant parts of the responses.
